# Nine Months of a Structured Multisport Program Improve Physical Fitness in Preschool Children: A Quasi-Experimental Study

**DOI:** 10.3390/ijerph17144935

**Published:** 2020-07-08

**Authors:** Boris Popović, Milan Cvetković, Draženka Mačak, Tijana Šćepanović, Nebojša Čokorilo, Aleksandra Belić, Nebojša Trajković, Slobodan Andrašić, Špela Bogataj

**Affiliations:** 1Faculty of Sport and Physical Education, University of Novi Sad, 21000 Novi Sad, Serbia; borispopovic0803@gmail.com (B.P.); cveksha@gmail.com (M.C.); macak.md@yahoo.com (D.M.); tijanascepanovic021@gmail.com (T.Š.); cokorilon@gmail.com (N.Č.); play.dance.studio@gmail.com (A.B.); nele_trajce@yahoo.com (N.T.); 2Faculty of Economics, University of Novi Sad, 24000 Subotica, Serbia; andrasicslobodan@yahoo.com; 3Faculty of Sport, University of Ljubljana, 1000 Ljubljana, Slovenia; 4Department of Nephrology, University Medical Centre, 1000 Ljubljana, Slovenia

**Keywords:** exercise training, physical fitness, preschool, physical activity

## Abstract

Research in preschool children that investigates the impact of different exercise interventions on physical fitness is limited. This pre–post study was aimed at determining if participation in a nine-month structured multisport program (MSG; n = 38) could enhance physical fitness components compared to a formal exercise program (control group (CG); n = 36) among preschool children. Physical fitness was assessed using standardized tests (the standing long jump, sit and reach, 20 m sprint, sit-ups for 30 s, bent-arm hang, medicine ball throw (MBT), grip strength, 4 × 10 m shuttle run, and 20 m shuttle run tests). The structured multisport program involved fundamental/gross and fine motor skills and ball game-based exercises twice a week. The control group was free of any programmed exercise except for the obligatory program in kindergartens. A mixed ANOVA demonstrated significant group-by-time interaction effects for the 4 × 10 m shuttle run, standing long jump, sit-ups, bent-arm hang, grip strength, and sit and reach tests (*p* < 0.05). There was no significant group-by-time interaction effect for the 20 m sprint test (*p* = 0.794) or for the 20 m shuttle run test (*p* = 0.549). Moreover, the MSG and CG performance in the MBT and 20 m shuttle run tests improved to a similar extent from pre- to post-test. Our results indicate that compared to the formal plan, the structured multisport program led to a sustained improvement in physical fitness in healthy 5-to-6-year old children.

## 1. Introduction

Physical fitness is an excellent indicator of health in children and adolescents [1] and a predictor of a healthier cardiovascular profile later in life [2]. Physical fitness also has a positive effect on a wide range of psychological indicators (depression, cognition, social and sports competence, and self-esteem) [3]. Moreover, preschool children who regularly participate in physical activity have long-term benefits for physical and psychological well-being [4]. Therefore, the consistent monitoring of the level of physical fitness of individuals should be a priority for public health [5]. Accordingly, exercise programs with the aim of improving physical fitness in children and, consequently, improving their health are needed [4]. Unfortunately, there are limited data on preschool-aged children.

There is clear evidence that many children do not meet most of the international physical activity recommendations, which is 60 min of moderate-to-vigorous physical activity every day [6]. Moreover, it was stated that children who spent more time in a childcare environment each day were less active than children who spent longer periods in outside settings [7]. Recent studies have reported that young children are sedentary for almost 50% of their time at childcare [8,9]. The importance of childcare time in supporting physical activity among preschoolers was confirmed by Vanderloo et al. [10]. The standardized physical activity lessons among kindergartens in Serbia consist of 30 min of organized physical activity, mainly containing traditional games. However, most kindergartens have no organized physical activities but only free play. Therefore, structured training programs can play a significant role, as they are considered to be an excellent means for improving and promoting motor and health-related fitness in early childhood [11]. The importance of structured and multicomponent training programs for preschool children was confirmed recently [12,13]. Roth et al. [12] have provided evidence that motor fitness performance in preschool children can be improved and maintained by an appropriate physical activity program in 4-to-5-year-old boys and girls. Similarly, Krneta et al. [13] found significant effects of additional kinesiological treatment on explosive strength (standing broad jump) and flexibility (seated straddle stretch) in preschool boys.

Early childhood is considered an ideal age period for the development of fundamental movement skills [14], as well as for the development of physical activity [15]. However, studies that have investigated the importance of physical fitness and the effects of different exercise programs on it in preschool children showed different results, mostly due to large variability in quality and the used methods. A novel meta-analysis (19 studies) showed that physical exercise, whether combined or not with diet, has a small effect on physical fitness in preschoolers [4]. Therefore, exercise programs with different activities and with vigorous intensity at young ages could be one of the solutions. The purpose of this study was to determine if participation in a nine-month structured multisport program could enhance fitness components among preschool children compared to those in a group that attended regular physical activities in kindergartens.

## 2. Materials and Methods 

### 2.1. Participants

The experimental protocol received approval from the institutional ethics committee from the Faculty of Sport and Physical Education, University of Novi Sad (Ref. No. 32/2017). Written informed consent was obtained from the parents of all the children who gave their assent to participate in the study. The flow of participants through the study is presented in Figure 1.

A total of 110 children from 5 to 6 years old and their parents were invited to take part in the study. Written informed consent was provided from the parents for the 101 eligible children. Two children had to be excluded as they did not meet the inclusion criteria (health reasons). Finally, the study was performed on a sample of 74 healthy preschool children aged 5 to 6 that were divided into two groups. The MSG (n = 38) consisted of children attending a structured multisport exercise program twice per week. The recruitment of children in the MSG started earlier with an invitation of all preschools from the local area to participate in a national project regarding novel exercise programs in preschool children. The CG (n = 36) was randomly chosen from the cluster of local kindergartens. Children of the control group were not informed about the existence of the intervention program in other preschools. 

### 2.2. Testing Procedures

The set of anthropometric variables and the set of physical fitness variables obtained by the use of the following measurements and tests were analyzed at the beginning of both treatments and after nine months. Body height was measured with a fixed anthropometer according to Martin (GPM Anthropometer 100; DKSH Switzerland Ltd., Zurich, Switzerland; ±0.1 cm), and body mass, with a digital balance (BC1000, Tanita, Japan; ±0.1 kg), following the guidelines proposed by the International Biological Program (IBP). 

We used the PREFIT and EUROFIT test battery to assess physical fitness in preschool children [16,17,18]. Cardiorespiratory fitness and muscular strength are proposed as powerful health markers; therefore, this study defined the modified 20 m shuttle run, standing long jump, and grip strength as primary outcomes. Secondary study outcomes included muscle endurance tests (sit ups for 30 s and bent-arm hang), the medicine ball throw, the 20 m sprint, the 4 × 10 m SRT (shuttle run test), and sit and reach tests. 

#### 2.2.1. Primary Outcomes

The **modified 20 m shuttle run test** assessed cardiorespiratory fitness (CRF) [16]. Briefly, children were required to run back and forth on a 20 m course with an audio signal. The test was finished when the child failed to reach the end lines concurrent with the audio signal on two consecutive occasions or when the child stopped because of exhaustion. Some adaptations of the original test were made due to young age of the preschoolers by decreasing the initial speed (i.e., 6.5 km/h instead of the original 8.5 km/h) and by having two assistants running with a reduced group of children (e.g., 4–8 preschoolers of the same age) in order to provide an adequate pace [19]. The test results were expressed as the number of laps completed. 

The **standing long jump**. Explosive strength was measured by the standing long jump test [16]. The participants were asked to swing their arms and jump forward onto a carpet, which was marked in cm. Children performed three jumps, and the result was the length of the jump in cm. The best of these attempts was recorded.

**Grip strength** to assess upper-limb muscular strength was measured with a baseline pneumatic squeeze bulb dynamometer (Baseline, USA). The participants were positioned in a sitting position with an elbow on the desk. Two successive bilateral grip strength measurements were recorded. The dominant hand was measured first. The best value of the two trials for each hand was chosen. A 30-s rest period between each measurement was given to prevent fatigue. The dynamometer was supported slightly by the examiner to prevent any accidental falls. All the tests were performed by the same examiner. Reliability in preschool children was documented previously [20].

#### 2.2.2. Secondary Outcomes

The **medicine ball throw (MBT)** to evaluate upper-limb power [18]. Participants had to push a medicine ball (1 kg) with two hands as far as possible. The starting position was with the feet parallel to each other and shoulder-width apart, with the ball held against the chest. The test item score (better of two attempts) was the distance achieved in cm.

The **bent-arm hang** to assess upper limb muscular endurance [17]. The child under-gripped the bar and held the pull-up as long as he/she could (chin above the bar). The result was the time of the hold measured in seconds.

The **sit-ups for 30 s test** was used for measuring the muscular endurance of the trunk [17]. The child lay on his/her back with his/her knees bent and hands clasped behind their neck. He/she rose into a seated position and returned to the starting position. The result was the number of correctly performed sit-ups in 30 s.

The **20 m sprint** to assess speed and anaerobic power. At the command “GO”, the child that stood behind the start line had to run 20 m as fast as he/she could to the end of the track (20 m). The score was the time of running, measured in seconds [18].

The **4 × 10 m SRT** assessed speed-agility. The children had to run and turn at maximum speed between two parallel lines (10 m apart) drawn on the floor, covering a distance of 40 m. To make this easier, two assistants were positioned on both ends, and the participants had to touch their hands (placed behind the line) and go back at maximum speed. The better of two attempts was recorded (seconds) [16].

The **sit and reach test** assessed flexibility [17]. The participants were instructed to sit on the floor barefoot, with their legs together, knees fully extended. Participants were then asked to bend forward slowly and use their arms to push the edge of a moveable board as far as possible without bending their knees. The measuring stick on the device had the zero mark at 25 cm before the feet, and the result was recorded directly from the meter on the device. The task was repeated twice, and the better of two attempts was recorded in cm.

The reliability of the physical fitness tests was previously determined in the same age group. Good reliability coefficients were obtained for all the tests, with the intraclass correlation (ICC) ranging from 0.86 to 0.97.

### 2.3. The Structured Multisport Exercise Program 

The structured multisport exercise program lasted nine months and was conducted indoors in a fully equipped gym two days per week for approximately 60 min (Table 1). The sessions comprised multiple sports activities and exercises designed and supervised by exercise science researchers and led by trained physical education teachers. Each week’s activities focused on a skill or group of skills from one of the three gross motor skill categories: stability (trunk strength), locomotor (running, hopping, and skipping), or manipulation (ball skills). Additionally, children were introduced every week to the most important elements from the team and individual sports. Early in each week, children were introduced to motor skills, and movement concepts were added at the end of the week. Later in the program, skill patterns were incorporated into activities. We used the Borg rate of perceived exertion (RPE) scale (1–10) and asked each participant to provide their overall RPE for the session. It was shown previously that children, even as young as 5–6 years of age, can become quite adept at rating their perceived exertion [21]. The structured multisport exercise programs were organized in the form of frontal, group work but mostly as a circuit training (polygon) or repetitive (station) training. If the exercise was performed within the polygon, children were to do the task and move to the next station. Obstacles consisted of various gymnastic apparatus and props for solving motor problems made up of gymnastic and athletic exercises, as well as elementary games. Station training was made up of repeating the same exercises more than once on one station (e.g., one gymnastic apparatus). In this case, the exercise was repeated. In order to increase the intensity of the exercises and reduce the waiting for the apparatus for exercising, some exercises with additional and complementary tasks were added (e.g., exercises for strengthening and stretching, as well as exercises on supplementary apparatus).

Structure of training: (I) Warm-up—15 min. Various movements with changeable speed, exercises that correct and prevent flat-feet, stretching, corrective and preventive exercises to improve bad posture, and a proper sense of good performance. (II) Main part—40 min. Revision and practice of previous skills, teaching and practicing new skills, competitive practice, and conditioning. (III) Cool-down part—5 min. Stretching, coaching comments, and conversation. Every part of the training lesson was run with a positive and warm, friendly mood, with proper music (particularly in the introduction and preparation). 

The control group attended regular activities in kindergartens that included different means of exercising, learning methods, and exercising itself, the purpose of which was to fulfill the requirements of the formal plan and program of preschool institutions. The regular activities in kindergartens were conducted in a small kindergarten gymnasium, typical of all gymnasiums in preschool institutions. The regular activities were delivered by kindergarten teachers, who were non-experts in physical education. The gymnasium was modestly equipped with the most necessary props.

### 2.4. Statistical Analysis

Data were analyzed using SPSS v. 20.0 and are presented as mean ± SD unless otherwise stated. A priori, the G*power 3.1 power analysis software [22] determined the minimum sample size (n = 52) given the critical F = 4.034, effect size = 0.2, *p* = 0.05, 1–β = 0.8, groups and time points = 2, and corr = 0.5. There were no outliers in the data, as assessed by the inspection of a boxplot. If a Shapiro–Wilk test did not fail to reject normality, log-transformed data were analyzed, but original data are reported for the sake of clarity. The Levene’s and Box’s tests failed to reject the homogeneity of the variances and covariance matrices, respectively. A 2 (MSG and CG) x 2 (pre- and post-test) mixed-model ANOVA was used to evaluate the intervention-induced effects on studied physical fitness measures, and thus, the primary tested hypothesis was the group-by-time interaction effects. Moreover, we tested the simple main effect of time to show changes from before and after the tests in MSG and CG, and the simple main effect of the group to show differences between MSG and CG before and after the tests. The Bonferroni adjusted *p*-values for multiple comparisons were determined. Eta squared (ŋ^2^) and partial eta squared (${ŋ}_{p}^{2}$) are reported as measures of the effect size for the simple main effects and interaction effect, respectively, and defined as small (0.01), medium (0.06), and large (0.14) [23]. The level of significance was set at *p* ≤ 0.05. 

## 3. Results

Participants’ demographic information is presented in Table 2. Moreover, information concerning the adherence, drop out, and injuries are presented in Figure 1. There were no injuries during the structured multisport exercise program or regular activities in kindergartens that could have caused exclusion from the study.

Table 3 presents mixed ANOVA outcomes. The simple main effect of the group showed that most of the physical fitness measures on average differed between the MSG and CG pre-test (20 m sprint, 4 × 10 m SRT, MBT, bent-arm hang, grip strength, and modified 20 m shuttle run) and post-test (20 m sprint, standing long jump, sit-ups for 30 s, bent-arm hang, and modified 20 m shuttle run). Hence, the intervention-induced effects were examined by testing the group-by-time interaction effects to transcend the unbalanced groups. 

### 3.1. Primary Outcomes

A simple main effect of time showed significant improvements in the MSG and CG in all the primary outcomes from pre- to post-test. However, a group-by-time interaction effect showed a significantly higher increase in the mean length of the standing long jump in the MSG than in the CG (*p* = 0.032). By contrast, a significantly greater increase in the mean grip strength was observed in the CG than in the MSG (*p* = 0.046).

### 3.2. Secondary Outcomes

The MSG significantly improved in all secondary outcomes from pre- to post-test, as showed by a simple main effect of time. Regarding the CG, the significant improvements were not found for the sit-ups 30 s (*p* = 0.938), bent-arm hang (*p* = 0.127), and sit and reach (*p* = 0.179). A group-by-time interaction showed that MSG had significantly larger increases of mean sit-ups (*p* = 0.015) and bent-arm hang (*p* < 0.0005) performances, and larger decreases of mean time spent to finish the 4 × 10 m SRT (*p* < 0.0005), as compared to CG. MBT test performance significantly improved to a similar extent for both MSG and CG (*p* = 0.066). Table 3 presents detailed information on the results of the analysis.

## 4. Discussion

Physical fitness is one of the major indicators of health in children and adolescents. The present study assessed the impact of a 9-month structured multisport program on physical fitness in preschool children. The major findings of this study were that structured multisport training improved physical fitness, with positive training effects on strength, as well as on explosive power, speed-agility, CRF, and flexibility. In the case of the control group, the changes were similar, except for muscle endurance and flexibility, where the changes were not significant. Increases in height, weight, and BMI are usual for this age group. 

The experimental program consisted of a 60 min of physical activity intervention two times a week designed and supervised with experts and researchers who are Ph.D. holders in sport and exercise science. Improvement in physical fitness was based on statistically relevant benefits for the multisport group in strength (grip strength, bent-arm hang, and sit-ups), explosive leg strength (standing long jump), speed-agility, CRF, and flexibility. A recent meta-analysis [4] confirmed that physical activity interventions in preschool children could induce positive changes in cardiorespiratory fitness, lower-body muscular strength, and speed-agility. However, the authors stated that the changes were related to body composition in preschoolers and recommend interventions with vigorous intensity. The biggest improvements in the MSG were documented for the standing long jump (ES = 0.32) and speed-agility (0.67). Similarly to our study, Roth et al. [12] have provided evidence that speed-agility performance in preschool children can be improved and maintained by an appropriate physical activity program in 4-to-5-year-old boys and girls. Moreover, a recent study [24] in preschoolers showed that a play-based, non-competitive physical activity intervention induced significant improvements in muscular strength and motor fitness, which includes speed, agility, and the coordination of movements. The better improvements in musculoskeletal fitness, speed-agility, and flexibility in the current study were mainly due to the fact that the structured multisport program was based on motor coordination, stability, and locomotor and object control skills and activities. The emphasis in the experimental program was on the energy aspect of physical exercises, classes, and games. Moreover, the program was designed according to several recommendations [25,26] and based on strength, the strengthening of all body segments, and perceptual-motor training. Therefore, it was logical that the most significant improvement was in strength, explosive strength, and flexibility. However, it should be noted that higher grip strength improvements in the CG than in the MSG were expected, bearing in mind the large differences at the beginning of the program. The MSG had high results in the initial measurements, which may have significantly contributed to the smaller improvements in the final measurement. This was confirmed in the study by Trajković et al. [27] that showed lower grip strength values in the children from the CG in the initial measurement. 

Cardiorespiratory fitness measured using the modified 20 m shuttle test was enhanced in both the experimental (ES = 0.11) and control groups (ES = 0.25) with no significant differences between the groups. The reason for the no significant differences between the groups is mainly the great differences between the groups at the beginning of the program. Children from the MSG already had high results for CRF because they were already enrolled in the program. On the contrary, children from the control group (17.67) had lower values according to the norms (25–30) for this age group [19,28]. Therefore, similar improvements in CRF compared to those in the CG were logical. Moreover, the children in the control group were enrolled in regular kindergarten activities along with free play for most of the leisure time. Two recent meta-analyses showed that exercise interventions could improve CRF; however, the reported magnitude of the effects was small [4,29]. One study in preschool children failed to obtain significant improvements in CRF following a three 60 min sessions/week physical activity intervention [24]. The authors showed improvements in girls only, which was due to the smaller baseline values for their CRF compared to that of boys. 

Several authors point out that cardiorespiratory fitness is one of the most important components of health-related fitness [1,26,30]. Additionally, these authors point out that the development of CRF is more stimulated by vigorous-intensity and outdoor activities. Therefore, we could assume that for greater progress, more outdoor activities are needed, which are carried out on open fields or on playgrounds. However, due to the fact that young children spend the majority of their time in childcare, intervention in the daycare setting may be an ideal environment for the promotion of physical fitness and physical activity [6]. This was confirmed in the current study, which showed great improvements in physical fitness from adding just two sessions per week to the obligatory childcare activities. Therefore, implementing these programs in everyday activities would be of great importance, especially because of the limited number of published exercise interventions conducted with preschool-age children. 

Some study limitations should be noted. We did not follow the children’s daily unorganized activities and inactivity that could have possibly influenced the physical fitness and development of abilities. We know that children spend approximately 50% of their time sitting during childcare [8]. Therefore, examining physical activity throughout the entire day would give us a clearer picture of the possible mediators in the current study. The relatively small sample size should also be considered as a limitation. Moreover, this was a quasi-experimental study. Bigger and cluster-randomized studies are needed to determine the true effects. Nevertheless, this study provides evidence regarding the effectiveness of structured multisport programs for the optimal development of young children’s physical fitness. 

## 5. Conclusions

Our results indicate that the structured multisport program led to a sustained improvement in physical fitness in healthy 5-to-6-year old children. We provided evidence that the program could be implemented in indoor facilities and that a frequency of only two times per week was shown to be enough for significant improvements compared to the control group. These and similar findings could be of public health importance because physical fitness as early as in childhood is associated with a healthier cardiovascular profile later in life. Further research with longer interventions is needed in order to see if fitness levels tend to persist from childhood through adolescence to adulthood, which might have great cardiometabolic benefits.

## Figures and Tables

**Figure 1 ijerph-17-04935-f001:**
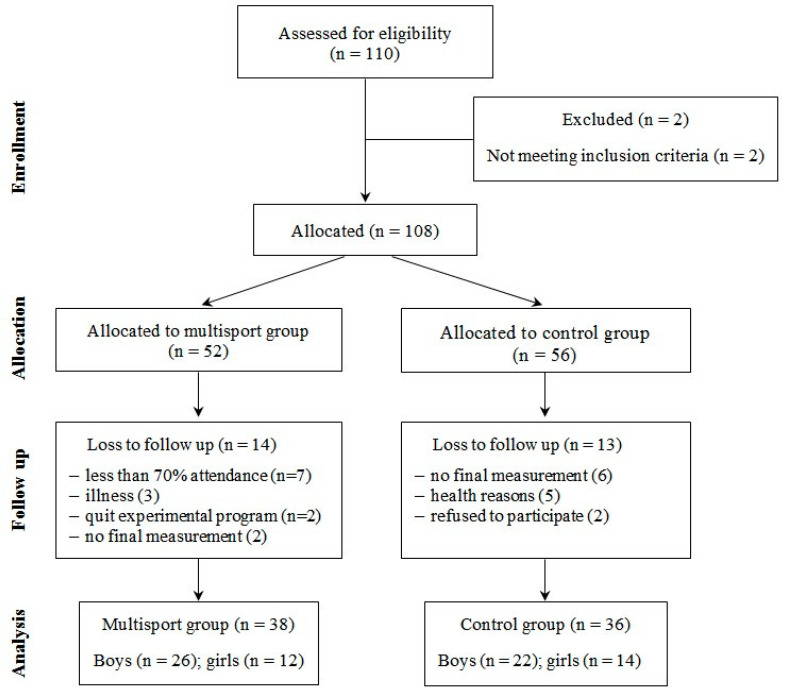
Flow diagram of participant enrolment, randomized group allocation, and final analysis.

**Table 1 ijerph-17-04935-t001:** The structured multisport exercise program.

Duration	Organization	Volume	Frequency	Intensity
9 months	Frontal work, group work, work with stations, polygon/circuit work, and obstacle courses	~60 min per session	2 times a week	According to external signs (sweat, blush, spontaneous breaks); RPE 5–8 (moderate to vigorous intensity)

Abbreviations: RPE, rate of perceived exertion.

**Table 2 ijerph-17-04935-t002:** Bodyweight status pre- and post-intervention.

	MSG		CG	
	Baseline	Post	Baseline	Post
Age (years)	5.03 ± 0.29		5.36 ± 0.22	
Body height (cm)	114.15 ± 5.27	118.44 ± 5.28	113.79 ± 4.87	116.14 ± 5.32
Body mass (kg)	20.38 ± 3.94	22.32 ± 3.05	19.80 ± 3.42	20.60 ± 2.27
BMI	16.40 ± 1.70	16.26 ± 1.28	15.42 ± 1.18	15.28 ± 1.30

Note: All data are presented as mean ± SD.

**Table 3 ijerph-17-04935-t003:** Differences between MSG (n = 38) and CG (n = 36) in physical performance measures from pre- to post-test.

		Pre-Test	Post-Test	ES	A Group-by-Time Interaction Effect
Modified 20 m shuttle run test (freq.)					
	MSG	26.82 ± 6.81 †	30.23 ± 10.38 † *	0.11	F_(1, 72)_ = 0.363; *p* = 0.549; ${ŋ}_{p}^{2}$=0.006; 1-β = 0.09
	CG	17.67 ± 6.29	22.08 ± 5.71 **	0.25	
Standing long jump (cm)					
	MSG	116.38 ± 18.33	126.60 ± 15.65 ^†^ **	0.32	F_(1, 72)_ = 4.799; *p* = 0.032; ${ŋ}_{p}^{2}$=0.062; 1-β = 0.58
	CG	113.66 ± 16.03	117.97 ± 16.26 *	0.06	
Grip strength (PSI)					
	MSG	4.75 ± 1.08 †	5.08 ± 1.07 *	0.07	F_(1, 72)_ = 4.119; *p* = 0.046; ${ŋ}_{p}^{2}$=0.054; 1-β = 0.517
	CG	4.05 ± 1.38	4.81 ± 1.49 **	0.24	
MBT (cm)					
	MSG	241.86 ± 51.23 †	255.62 ± 44.74 *	0.13	F_(1, 72)_ = 3.473, *p* = 0.066; ${ŋ}_{p}^{2}$=0.046; 1-β = 0.45
	CG	216.41 ± 43.19	242.31 ± 47.51 **	0.28	
Sit-ups for 30 s (freq.)					
	MSG	11.19 ± 5.53	13.90 ± 4.28 † **	0.17	F_(1, 72)_ = 6.237; *p* = 0.015; ${ŋ}_{p}^{2}$=0.080; 1-β = 0.61
	CG	11.31 ± 4.35	11.38 ± 3.98	0.00	
Bent-arm hang (s)					
	MSG	17.47 ± 18.40 †	26.68 ± 23.72 † **	0.22	F_(1, 72)_ = 17.222, *p* < 0.0005; ${ŋ}_{p}^{2}$=0.193; 1-β = 0.98
	CG	9.29 ± 10.59	5.70 ± 6.17	0.03	
20 m sprint (s) ^¥^					
	MSG	5.03 ± 0.48 †	4.84 ± 0.39 † **	0.16	F_(1, 72)_ = 0.069; *p* = 0.794;, ${ŋ}_{p}^{2}$=0.001; 1-β = 0.58
	CG	5.45 ± 0.46	5.29 ± 0.41 *	0.10	
4 × 10 m SRT (s) ^¥^					
	MSG	17.01 ± 1.21 †	15.18 ± 1.22 **	0.67	F_(1, 72)_ = 24.577; *p* < 0.0005; ${ŋ}_{p}^{2}$=0.281; 1-β = 0.998
	CG	15.55 ± 1.09	14.85 ± 1.09 **	0.23	
Sit and Reach (cm)					
	MSG	28.98 ± 4.27	32.45 ± 5.16 **	0.44	F_(1, 72)_ = 35.570; *p* < 0.0005; ${ŋ}_{p}^{2}$=0.331; 1-β = 1.0
	CG	30.88 ± 4.28	30.16 ± 4.75	0.03	

Values are means ± SDs. Abbreviations: ^¥^, reverse scoring; freq., frequencies; PSI, pounds per square inch; MBT, medicine ball throw; A, mixed ANOVA outcomes; ES, eta squared (ŋ^2^) for pre- to post-test changes; F, F-test statistics; p, probability value; ${ŋ}_{p}^{2},$ partial eta squared; 1-β, (post-hoc) statistical power of the test; †, groups significantly different at *p* < 0.05 (the simple main effect of group); *, significant pre- to post-test changes at *p* < 0.05 (the simple main effect of time); **, significant pre- to post-test changes at *p* < 0.0005 (the simple main effect of time).

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
