# Peer review of "Nine Months of a Structured Multisport Program Improve Physical Fitness in Preschool Children: A Quasi-Experimental Study"

_ijerph, 2020, doi:10.3390/ijerph17144935_

Round 1

Reviewer 1 Report

The authors must improve table 3. It is not clear. I recommended to write the tests in a column at the left, and the ANOVA results in three columns, one for group effect, one for time effect, and the last one for the group x time interaction.

The authors must review the references paragraph, some errors are detected. 

Author Response

Reviewer 1

The authors must improve table 3. It is not clear. I recommended to write the tests in a column at the left, and the ANOVA results in three columns, one for group effect, one for time effect, and the last one for the group x time interaction.

Our response: Thank You. You are right; the Table 3 was unclear and sloppy. We denoted the simple main effect of group and time with symbols (†, *), while the interaction effect has been left intact. So please find the revised Table 3 in the new version of the manuscript. 

The authors must review the references paragraph, some errors are detected.

Our response: The reference list was corrected.

Reviewer 2 Report

This interesting study propose to compare the effects of a structured multisport program vs the formal plan and exercise program of preschool institutions, on physical fitness in young children (girls 5 to 6 years old). The authors emphasize the importance to improve physical fitness in childhood (because of a link with cardiorespiratory fitness in adults which is the main criterion of morbi-mortality). The strengths of the study: its duration, 9 months which allows to appreciate the improvements. The different physical components which are evaluated and which allow to draw up a “performance map” (speed, strength, flexibility… etc…). In order to improve the article and the understanding of the design/results/discussion, I have some comments:

-I understood than in kindergartens, children already have an exercise-training plan. Title suggestion: Nine Months of Structured Multisport Program Improve Musculoskeletal Fitness in Preschool Children

-Key-word: intervention and impact could be replace by exercise training and physical activity?

Abstract

The results part of the abstract could be shortening.

Line 16 to 19: please shorten and precise that the CG is an exercise program of preschool institutions. Suggestions: This longitudinal study aimed to determine if participation in a nine-months structured multisport program (MSG n=38) could enhance fitness components compared to the formal exercise program (control group CG=36) among preschool girls.

Line 23: A mixed ANOVA demonstrated a statistical significant (…)

Line 24: group-by-time interaction effects for run 4x10m (…)

Line 25: rephrased (p=0.0005, 0.032, 0.015, 0.0005, 0.046, 0.0005, respectively). Suggestion: (p<0.05).

Line 26: group-by-time interaction effect for the speed 20m test (p=0.794) neither for 20 m shuttle run test (p=0.549).

Line 26 to 30: please shorten. Suggestion “Moreover, MBT and shuttle-run-20m test performance improved to a similar extend in the two groups”.

Line 31: Compared to formal plan/ the structured multisport program

Introduction

-There is some of redundancy in the introduction. For example (but not only) the first two sentences are similar; line 55 to 57 could be synthetize.

-Can you explain “adult health” and “future health” in the introduction (and discussion part e.g line 214): do you refer to VO2max, cardiovascular risk factor, fat mass, metabolic diseases, cognitive decline? Please precise.

- is there a demonstrated link between cardiorespiratory fitness in childhood and VO2max in adults?

-What are the international physical activity recommendations for young children? to my knowledge, 60 minutes of moderate to high intensity physical activity per day is recommended. Please precise clearly.

- Can you describe in the introduction what is the formal plan and exercise program of preschool institutions? what are the physical activities generally proposed in kindergartens ?

-Line 46: please precise « other environments »

-Line 47 « structured » training program?

-Line 60: please precise “additional intervention”

-Line 63 “multisport program with …etc…”

Participants Section 2.1

-I’m surprised to see here that your study population is only girls (line 66): is there some sex and gender considerations in your paper? And/or in the topic of physical activity in preschool children ? why did you choose only girls? may a statement in the introduction could be relevant to clarify and then to discuss in the Discussion section ? Please, add this information (girls) also in the abstract line 18.

-Line 66: n=56 is different than n=74… please clarify

-Line 68-69: it is a results. please shorten e.g the average age of the children was between 5 to 6 years old.

-Line 68: I understand that it is not a randomized study? may you could clarify and explain it before describing your 2 groups. It is important regarding your baseline-test results differences between your two groups.

-I suggest you to rephrase the Participants section. Please could you separate the information of the consent form and Ethical committee from the results and the description of your groups (line 67, and line 70-73). E.g. first part of your Participants Section: Ethical committee and Consent Form; Second part of your Participants Section: descriptions of the 2 groups, randomisation ?, average age.

-Table 1 in the Results section.

Testing Procedures 2.2

-In addition to weight, body fat percentage, and hydration levels, the BC-1000 Body Composition Monitor provides measurements on muscle mass, metabolic age rating, bone mass, and visceral fat. I don’t see any datas of body composition expected the total body weight. Please explain.

-Line 83: Please could you add one/two references for these tests, they are all commonly used /appropriate with your population? May you could complete your line 83 with a short sentence (e.g The following test battery was used in physical fitness assessment and are commonly used with preschool children. ref xx,xx).

-Is it possible to organize the list of the test by “domain” like cardiorespiratory fitness (20m, Shuttle run 20m), strength (handgrip, Standing broad jump, xxx), flexibility (xxx, xxx)? It could help the lecturer to understand.

-Line 85: I understood that your population is only girls (please, delete “he”)

-Line 84 to 112. All this section is well described.  Some of tests do not have references (e.g 20 m dash, Standing broad jump, Sit and reach test, Bent arm hang, Sit-ups 30 s test; is that test are commonly used with preschool children?)

-Line 122: I supposed that ICC is Intraclass correlation?

2.3. Multisport exercise program

-Table 2: RPE 5-8. Is it on a 10 points scale? In the table caption, please precise for the lecturer that 5-8 is corresponding to very-light or light, moderate, heavy, maximal intensity?

-The multisport exercise group have their MS exercise program (2 days/week x 9 months) with sports teachers + the control treatment the others days described line 142 with the kindergarten teachers? It is not clear what are they doing the others days of the week.

Statistical analysis

-How sample size was determined?

-Defined pre-specified primary and secondary outcome measures.

-Who was blinded after assignment to interventions (participants, those assessing outcomes?)

Results section

-Please add your table 1 in this section. And the mean age in each group +/- SD.

-As recommended by CONSORT (http://www.consort-statement.org/), Add a flow diagram (for each group, number of pre-screening, losses and exclusions (with reasons) after groups allocation, n analysed, ..etc…  and a short section at the beginning of your Results section about adherence/drop out/injuries…etc…

-Table 3: Tcheck your n in table 3. (it is different from line 66).

-There are many tests and the authors did not choose a First outcome.

Personal commentary: is it relevant to construct standardized Z-scores for each test? and then, you should calculated a physical composite scores for different physical domains using your raw Z-scores (such as CRF, resistance, power, …etc…). It helps to simplify the results section/interpretation/discussion.

Discussion

-Line 207: you unfortunately did not measured the body composition

-Line 227: could you clarify how indoor/outdor exercises activities could influence musculoskeletal fitness rather than CRF.

-Line 234-235: you stated that there is no significant differences between groups in CRF improvements. Do you think that is only due to the maturation process? Your two groups did exercises training (formal training program in your CG is not a sedentary lifestyle). and it is well know that strength training is also used to improved CRF. May your results are diluted because of the two exercises interventions.

Limitation

-line 256-258 : do you have any references to describe clearly the siting time and physical activity time during free-time (unorganized activities) in children 5-6 years old ? it could be interesting to describe it shortly here.

-Can you generalised your results to all the children, included boys?

Conclusion

-Line 263: the structured multisport program

-Line 264: I recommend to precise that your study population is healthy girl 5 to 6 years old.

Author Response

Reviewer 2

This interesting study propose to compare the effects of a structured multisport program vs the formal plan and exercise program of preschool institutions, on physical fitness in young children (girls 5 to 6 years old). The authors emphasize the importance to improve physical fitness in childhood (because of a link with cardiorespiratory fitness in adults which is the main criterion of morbi-mortality). The strengths of the study: its duration, 9 months which allows to appreciate the improvements. The different physical components which are evaluated and which allow to draw up a “performance map” (speed, strength, flexibility… etc…). In order to improve the article and the understanding of the design/results/discussion, I have some comments:

Our response: Thank you for your detailed review of our manuscript and for providing some insightful and thought-provoking suggestions to strengthen our manuscript. We feel we have sufficient responses to each of your major concerns listed above, which are further detailed below, and hope that they alleviate the concerns you have regarding the approaches adopted in our manuscript.

-I understood than in kindergartens, children already have an exercise-training plan. Title suggestion: Nine Months of Structured Multisport Program Improve Musculoskeletal Fitness in Preschool Children

Our response: Thank you. We agree with you and we accepted your title suggestion:   Nine Months of Structured Multisport Program Improve Musculoskeletal Fitness in Preschool Children

-Key-word: intervention and impact could be replaced by exercise training and physical activity?

Our response: Thank you for your comment. We replaced the key-words as you suggested.

Abstract

The results part of the abstract could be shortening.

Line 16 to 19: please shorten and precise that the CG is an exercise program of preschool institutions. Suggestions: This longitudinal study aimed to determine if participation in a nine-months structured multisport program (MSG n=38) could enhance fitness components compared to the formal exercise program (control group CG=36) among preschool girls.

Our response: Thank you for your insight. We accepted your suggestions and change the sentences.

Line 23: A mixed ANOVA demonstrated a statistical significant (…)

Line 24: group-by-time interaction effects for run 4x10m (…)

Line 25: rephrased (p=0.0005, 0.032, 0.015, 0.0005, 0.046, 0.0005, respectively). Suggestion: (p<0.05).

Our response (line 23-25): Thank you, we took your suggestions into account.

Line 26: group-by-time interaction effect for the speed 20m test (p=0.794) neither for 20 m shuttle run test (p=0.549).

Our response: Thank you for your insight. We added missing articles (line 23, 26), and test (line 26-27).

Line 26 to 30: please shorten. Suggestion “Moreover, MBT and shuttle-run-20m test performance improved to a similar extend in the two groups”.

Our response: Thank you for your suggestion. It has been shorten likewise.

Line 31: Compared to formal plan/ the structured multisport program

Our response: Thank you for your comment. We accepted your suggestion and change the sentence.

Introduction

-There is some of redundancy in the introduction. For example (but not only) the first two sentences are similar; line 55 to 57 could be synthetize.

Our response: We agree with your suggestion. The sentences were revised accordingly.

-Can you explain “adult health” and “future health” in the introduction (and discussion part e.g line 214): do you refer to VO2max, cardiovascular risk factor, fat mass, metabolic diseases, cognitive decline? Please precise.

Our response: The requested terms were changed and now it is clearer. Moreover, according to reviewer 3 comment, the sentence from the discussion was moved to the conclusion part.

- is there a demonstrated link between cardiorespiratory fitness in childhood and VO2max in adults?

Our response: Yes, the reference was provided in the introduction but also in the discussion part.

-What are the international physical activity recommendations for young children? to my knowledge, 60 minutes of moderate to high intensity physical activity per day is recommended. Please precise clearly.

Our response:  We have added the sentence that precisely describes the most international PA recommendations.

- Can you describe in the introduction what is the formal plan and exercise program of preschool institutions? What are the physical activities generally proposed in kindergartens?

Our response: Most of the kindergardens in Serbia do not have conditions for organised PA classes as well as PA teachers. However, the following structure is the official kindergarden activity for those who fulfil the conditions. The control group continued their usual routine with two structured physical activity offered by the kindergarten teacher per week that last about 30 minute. The standardized physical activity lessons contained an initial 5 minutes warm-up introduction period, the main section (about 20 minutes) including the scheduled activities, and a cool-down section (about 5 minutes). The warm up period mostly consist of different type of locomotor skills such as walking, running, jumping, hooping activities. The main sections mostly consist of all kind of polygons, object control activities, many interesting traditional games, dances, aerobic, rhythmic activities that included music were implemented in this section. The kindergarten teachers in control groups were free to offer children all kind of unstructured activities during they usual routine day.

-Line 46: please precise « other environments »

Our response: The sentence was revised.

-Line 47 « structured » training program?

Our response: Thank you for your comment. We accepted your suggestion.

-Line 60: please precise “additional intervention”

Our response: The term was changed to more precise word. 

-Line 63 “multisport program with …etc…”

Our response: The aim was changed accordingly.

The change can be seen in line

Participants Section 2.1

-I’m surprised to see here that your study population is only girls (line 66): is there some sex and gender considerations in your paper? And/or in the topic of physical activity in preschool children? why did you choose only girls? may a statement in the introduction could be relevant to clarify and then to discuss in the Discussion section ? Please, add this information (girls) also in the abstract line 18.

Our response: We really apologize for the inconvenience. This is our mistake. Both sexes were included in this research, boys and girls

-Line 66: n=56 is different than n=74… please clarify

Our response: We apologize for the inconvenience, this was a typographical error. There were 74 children included in this study.

-Line 68-69: it is a results. please shorten e.g the average age of the children was between 5 to 6 years old.

Our response: Thank you for the insight. We removed the sentence, and added age in the first sentence of the paragraph and in Table 1.

-Line 68: I understand that it is not a randomized study? may you could clarify and explain it before describing your 2 groups. It is important regarding your baseline-test results differences between your two groups.

Our response: Unfortunately, the study was not a randomisation study. The experimental group was selected earlier (lines). The control group was randomly chosen from different kindergartens and was not aware of the experimental design in another group.

-I suggest you to rephrase the Participants section. Please could you separate the information of the consent form and Ethical committee from the results and the description of your groups (line 67, and line 70-73). E.g. first part of your Participants Section: Ethical committee and Consent Form; Second part of your Participants Section: descriptions of the 2 groups, randomisation ?, average age.

Our response: Thank you. We agree with you, therefore, we revised the paragraph accordingly.

-Table 1 in the Results section.

Our response: Please find it in the Results section.

Testing Procedures 2.2

-In addition to weight, body fat percentage, and hydration levels, the BC-1000 Body Composition Monitor provides measurements on muscle mass, metabolic age rating, bone mass, and visceral fat. I don’t see any datas of body composition expected the total body weight. Please explain.

Our response: Thank you for your comment. You are right; however, we did not aim to evaluate intervention effects on body composition, but only on physical fitness components. We just aimed to describe our sample of subjects.

-Line 83: Please could you add one/two references for these tests, they are all commonly used /appropriate with your population? May you could complete your line 83 with a short sentence (e.g The following test battery was used in physical fitness assessment and are commonly used with preschool children. ref xx,xx).

Our response: Thank you for your comment. We accepted your suggestion and complete the sentence with three references.

-Is it possible to organize the list of the test by “domain” like cardiorespiratory fitness (20m, Shuttle run 20m), strength (handgrip, Standing broad jump, xxx), flexibility (xxx, xxx)? It could help the lecturer to understand.

Our response: Thank you for your insight. We agree with you, therefore, we revised and reorganized the list of the tests accordingly your suggestions. Furthermore, we clarify some test protocol and add missing test – the medicine ball throw.

-Line 85: I understood that your population is only girls (please, delete “he”)

Our response: As we mentioned above, both sexes were included in this research.

-Line 84 to 112. All this section is well described.  Some of tests do not have references (e.g 20 m dash, Standing broad jump, Sit and reach test, Bent arm hang, Sit-ups 30 s test; is that test are commonly used with preschool children?)

Our response: Thank you for your insight. We added references to all tests. Yes, all these tests are commonly used with preschool children and some of them even in younger age (3-5 years).

-Line 122: I supposed that ICC is Intraclass correlation?

Our response: Thank you for your comment. You are right, we added the term name.

2.3. Multisport exercise program

-Table 2: RPE 5-8. Is it on a 10 points scale? In the table caption, please precise for the lecturer that 5-8 is corresponding to very-light or light, moderate, heavy, maximal intensity?

Our response: Thank you for the comment. We added the training intensity in the table caption and also added explanation in the text.

-The multisport exercise group have their MS exercise program (2 days/week x 9 months) with sports teachers + the control treatment the others days described line 142 with the kindergarten teachers? It is not clear what are they doing the others days of the week.

Our response: Thank you for the insight. Unfortunately, we did not control leisure time of both groups.

Statistical analysis

-How sample size was determined?

Our response: We used G*power to estimate the total sample size using the following parameters (critical F=4.034, Effect size f=0.2, p=0.05, 1-β=0.8, groups and time points=2, r=0.5), and a total of 52 subjects were estimated to ensure above parameters. However, we included all available children (n=108) who fit the study criteria to overcome possible poor participants adherence to study, because of the long-lasting experiment: 9 months, that happened, indeed.

-Defined pre-specified primary and secondary outcome measures.

Our response: Primary outcome measures are 20m shuttle run, standing long jump, handgrip strength, because cardiorespiratory fitness and strength are proposed as powerful health markers  (DOI: 10.1038/sj.ijo.0803774). Secondary outcome measures are muscle endurance tests (sit ups 30s and bent arm hang), the medicine ball throw, 20m dash and sit and reach test and 4x10m shuttle run test.

-Who was blinded after assignment to interventions (participants, those assessing outcomes?)

Our response: This was not a blinded study. However, children of the control group were not informed about the existence of the intervention program in other preschools.

Results section

-Please add your table 1 in this section. And the mean age in each group +/- SD.

Our response: Please find the revised Table 2 (1) including age in the new version of the manuscript

-As recommended by CONSORT (http://www.consort-statement.org/), Add a flow diagram (for each group, number of pre-screening, losses and exclusions (with reasons) after groups allocation, n analysed, ..etc…  and a short section at the beginning of your Results section about adherence/drop out/injuries…etc…

Our response: Thank you for your insight. We add a suggested section at the beginning of Results section. Furthermore, please find flow chart diagram in the Results section.

-Table 3: Check your n in table 3. (it is different from line 66).

Our response: Please accept our apology, we made this by mistake.

-There are many tests and the authors did not choose a First outcome.

Our response: We agree with you that Table 3 was unclear and sloppy, so please find the revised Table 3 in the new version of the manuscript. 

Personal commentary: is it relevant to construct standardized Z-scores for each test? and then, you should calculated a physical composite scores for different physical domains using your raw Z-scores (such as CRF, resistance, power, …etc…). It helps to simplify the results section/interpretation/discussion.

Our response: Thank you for your insight. We agree with you that analysing related variables as composite score has certain advantage over raw scores, e.g. it gives us opportunity for thorough understanding of intervention-induced effect on each PF component. However, there are also  consequences of combining related variables into a composite variable (doi:10.1097/NNR.0b013e3182741948). They suggested to create composite variables during the planning stage and before any testing with outside variables. It should be also carefully planned which method will be used to create composite score (z scores, simple averaging, weighted averaging, meaningful grouping). We found that analysing raw scores tends to be most represented in the current literature. In our future work, we will certainly account for your suggestion.

Discussion

-Line 207: you unfortunately did not measured the body composition

Our response: We apologize for the inconvenience, we deleted the sentence.

-Line 227: could you clarify how indoor/outdoor exercises activities could influence musculoskeletal fitness rather than CRF.

Our response: Yes, please find the explanation in the new version of the manuscript (lines 280-293).

-Line 234-235: you stated that there is no significant differences between groups in CRF improvements. Do you think that is only due to the maturation process? Your two groups did exercises training (formal training program in your CG is not a sedentary lifestyle). and it is well know that strength training is also used to improved CRF. May your results are diluted because of the two exercises interventions.

Our response: We agree with your suggestion. The main reason for the results were significant differences in the initial measurement, with MSG showing higher results for CRF compared top CG. Therefore, smaller improvements were somewhat expected. This was explained in the discussion.

Limitation

-line 256-258 : do you have any references to describe clearly the siting time and physical activity time during free-time (unorganized activities) in children 5-6 years old ? it could be interesting to describe it shortly here.

Our response: We have added the sentences regarding the physical activity and sedentary time in the limitation paragraph.

-Can you generalised your results to all the children, included boys?

Our response: We have already mention mistake regarding participants characteristics and that boys were also included in the study. Sorry again for that mistake.

Conclusion

-Line 263: the structured multisport program

Our response: Thank you for your comment. We revised the sentence.

-Line 264: I recommend to precise that your study population is healthy girl 5 to 6 years old.

Our response: Our study included healthy 5 to 6 years old boys and girls.

Reviewer 3 Report

Abstract

  • First line: impact OF, not ON
  • Around line 27: “to a similar EXTENT”, not EXTEND

Introduction

  • Some sentences are not sound in English and should be reviewed so to make the text clearer
  • I am not a supporter of long introductions; however, I perceived this as being excessively short. Science should focus on being concrete and efficient, yet this section leaves out many questions and requires to be worked on and expanded, in my opinion. Some of the topics that, in my opinion, should be explained better:
    • What is physical fitness in pre-school childhood? What are the components? Toddlers’ physiology differs than kids’, adolescents’, and adults’ one. Perhaps it would be of interest for readers to know what toddlers’ physical fitness includes, and what research on the topic has done previously. Taking one by one the components of toddlers’ PF, what studies can be highlighted?
    • Sports in toddlers: as the authors are proposing a multi-sport program, what are previous experiences and studies on toddlers’ and sports? Even if non-multisport research, it is interesting to understand the background of sports at the age of the sample, so to justify the choice.
    • The authors should also develop the last part of the introduction, leading to the objectives. How are we supposed to understand and accept the intervention proposed by the authors, if we have no more than one sentence and their non-supported statement to foster it? I mean, reading from line 59 (A novel meta-analysis…) on, it is not really clear why “therefore, exercise programs with different activities” could be the solution. Based on what? Is there any study, even not directly related to the topic, that supports this idea? The way the topic is presented, it sounds more like “we want to try and see what happens” rather than a statement built on solid reasoning/background
  • Lines 42-43: who supports the idea that there is a need of exercise programs? No citation is given, and the information in the introduction does not clarify this issue
  • The authors say that: data is limited on programs for PF and health in preschool (lines 43-44, not supported by citation); there are several studies on exercise programs for fitness in preschool (lines 49-50, followed by two studies); there are limited studies investigating exercise programs in preschool (no citation). This is very confusing: several and limited are almost opposed concepts. Also, if studies are limited, what is the focus of the 19 studies in the meta-analysis reported? 19 studies perhaps are not many, yet it is already a body of research. Why not reporting the direct sources and what they found in those studies, rather than only mentioning this meta-analysis? How does the meta-analysis match the previous statements on limited/several studies?
  • Additionally, I would also suggest working on the logical sequence of topics presented so to give them order. Currently, the authors discuss (a) the importance of physical fitness in youth; (b) they declare there is a need for exercise programs to improve PF, with no supporting reference; (c) childcare importance for supporting active living; (d) they declare there are several studies on PA programs in pre-school (which contradicts their previous statement on the lack of data on preschool); (e) they switch to early childhood as an important age, mentioning that there is a lack of studies on it (few lines above, they stated the opposite – “several studies…” etc.); (f) finally, they mention that a meta-analysis showed that exercising has small effect on PF in preschoolers…I feel it is quite chaotic and difficult to follow, as well as contradicting itself in several points

Materials and Methods

Participants:

  • the sample size stated here (56) is inconsistent with that in the abstract (74) and with the sum of EG (38) and CG (36). Given that no information is provided about any participant dropping off, I assume it is a typo and needs to be corrected
  • Perhaps saying that they “volunteered” is not fully correct, as they are aged 5-6. It is quite difficult for me to imagine a 5-year-old kid saying he/she accepts to volunteer for a study, just as much as I find hard to imagine any kid of that age refusing to participate in some fun activity. I might be wrong, but I believe it would be more precise to say “whose parents gave permission” (as stated later) or similar
  • Were all kids from one kindergarten? From more? If from more, were there differences among them (rural/urban, socioeconomic level of the neighborhood, facilities available, etc.)? Those could be variables affecting the results, especially for the CG. Was the CG sample selected from the same kindergarten/s? Were they selected proportionally? A bit more information here would be helpful
  • Table 1 is misplaced, why presenting results on pre-post anthropometric measures in the “participants” section?

Testing procedures

  • What is the 20m dash testing? Speed?
  • Can the authors please report the reference/s for the validity data they mention in the section, please? It would be even better if the authors could report validity data in samples of the same age for each of the test separately (adding reference), rather than summing it up at the end with no cited work to support it.

Multisport program

  • Line 126: I think the authors meant “trained” rather than “educated”
  • I am a bit confused, perhaps because the authors use the word “treatment” for both EG and CG. So, the EG was assisted by trained PE teachers. On line 145, the authors say that “the treatment” was conducted by non-experts. I assume they refer to CG, therefore, I would suggest not to use the same term (treatment) for both groups so to reduce confusion. “Treatment” is typically associated with an intervention (EG), whereas CG simply followed the regular activities they were supposed to do (no treatment).
  • Is the program based on any literature? Is there any specific reason why it was built this way? Perhaps background literature, experts’ feedback, etc.? It would be great to understand the reasoning in this; maybe, if the authors can improve the introduction and justify better the need for implementing multiactivity programs, this would be come unnecessary here
  • What is the difference between “work with stations” and “circuit work”? Since the authors do not explain it in the text but mention it in the table, it would be nice to have more information (at least personally, for me to understand)

Statistical analysis

  • Please report the reference work for the threshold values of η2

Results

  • Please correct the word “extend”. It is, in fact, “extent”
  • Information in table 3 should be placed in a more standard way. There is too much text, whereas tables are made to avoid this. Also, the format is not consistent (for instance, why is there a border under “Speed”, yet not under any other test?)

Discussion

  • Why is BMI discussed? It seems to be not included anywhere except when it is reported as a table under “participants”. Either the authors include BMI (or body shape) as one of the objectives, they show the data in the results, they analyze the data as well, etc., or they should remove the information. Other would be if they had focused on health-related physical fitness, which usually includes body composition/shape. However, the flaws of the introduction do not help in this case. Had we had a better presentation of the research, perhaps the inclusion of BMI would be understood better
  • So, were the instructors trained PE teachers or PhD holders with a focus on sports and sciences? Or both? The information is not consistent, please unify it.
  • Lines 213-214: this sentence sounds more fitting in the conclusions rather than in the middle of the discussion
  • Line 246: I think “speculate” is not a proper term to use. I would rather use “assume”, or “postulate”, “theorize”, etc.
  • Lines 247-249: again, this would be more a “conclusion” sentence. However, the authors should not go beyond the boundaries of their study: they have not studied outdoors activities, and they only theorize that the lack of growth in CRF is due to the indoor nature of their proposed program. Therefore, reaching the conclusion that outdoor activity should be proposed cannot be justified
  • Why, in the authors’ opinion, has grip strength increased more in CG? This should also be discussed, I believe
  • The nature of the study is experimental. In the specific case, I am not sure about the meaningfulness of the results for a main reason: adding two hours of sports per week in kids’ routine can only improve their fitness compared to those who do not have such possibility. Especially in kids, whose development is triggered by several activities, I consider this a limitation. I think it is logical that if a kid can play 120 minutes more per week for nine months, compared to those who do not add that activity time, the fitness development is faster. Different would be if there had been another EG with a different set of activities to compare the multisport program with. So, all in all, what we can really say here is that extra 120 minutes/week of exercising per week seems to be better than “no extra activity”…
  • The above stated is made even more difficult considering that: the extra 120 minutes of activities did not improve CRF more than it did in CG, which did not have those 120-minute activities. The conclusion, basically, is that if a kid plays multisports 2h per week, and another does not do anything active in those 2h, CRF will increase similarly (there are plenty of mediating variables that I am not considering here, this is just for the purpose of clarifying my thought). So are multisports really good? Similar logical thought can be applied to grip strength, which increased more in kids who did not do anything rather than those who played multisports…the only thing I can think about is whether the kids in EG were not following the regular exercise time (kindergarten requirements, lines 142-144)…? No mention is given in the text, nor what requirements are applied to kindergarten in the community in which the study was carried out (2h/week? 1h/week?). Perhaps being more precise in describing these aspects would help better understand the meaningfulness of the results, regardless of their magnitude and statistical significance
  • While the discussion could delve more on potential causes of the results found, it is acceptable. However, in my opinion, there are several concepts that belong to the conclusions.

Final comment:

  • After reading the text, the definition “longitudinal study” given in the abstract is not correct. This is a pre-post study lasting 9 months. Not only the duration of a study defines the design, but also frequency of measures etc. Longitudinal studies suppose tracking the same sample over a long time with repeated measures, which is not the case. I suggest the authors remove this definition as it is misleading for readers
  • A moderate review of the language is suggested

Author Response

Reviewer 3

Thank you for your detailed review of our manuscript and for providing some insightful and thought-provoking suggestions to strengthen our manuscript. We feel we have sufficient responses to each of your major concerns listed above, which are further detailed below, and hope that they alleviate the concerns you have regarding the approaches adopted in our manuscript.

Abstract

First line: impact OF, not ON

Around line 27: “to a similar EXTENT”, not EXTEND

Our response: We apologize, we corrected those mistakes.

Introduction

Some sentences are not sound in English and should be reviewed so to make the text clearer

Our response: Thank you for point out this limitation. We have used Grammarly and native speaker has edited the manuscript.

I am not a supporter of long introductions; however, I perceived this as being excessively short. Science should focus on being concrete and efficient, yet this section leaves out many questions and requires to be worked on and expanded, in my opinion. Some of the topics that, in my opinion, should be explained better:

Our response: Thank you for your suggestion. We have added some additional explanation. However, research regarding the preschool children and interventions are really scarce and we did our best to introduce the importance of interventions as well as multisport activities in young children.  We hope that you will find our introduction satisfactory for further process.

What is physical fitness in pre-school childhood? What are the components? Toddlers’ physiology differs than kids’, adolescents’, and adults’ one. Perhaps it would be of interest for readers to know what toddlers’ physical fitness includes, and what research on the topic has done previously. Taking one by one the components of toddlers’ PF, what studies can be highlighted?

Our response: We have focused on preschool children and studies that are published by the top researchers in this area. We hope that we have improved our introduction according to your important suggestions and the suggestions of other reviewers regarding the importance of PF in preschool children.

Sports in toddlers: as the authors are proposing a multi-sport program, what are previous experiences and studies on toddlers’ and sports? Even if non-multisport research, it is interesting to understand the background of sports at the age of the sample, so to justify the choice.

Our response: We have mentioned two important studies regarding the importance of multisport activities in preschool children. Concept of early specialisation and sport in toddlers is very interesting but very hard to prove scientifically. The program we have proposed was based on authors personal experience working with preschool children for past several years. Moreover, mentioned studies also stated the importance of multiactivities in younger children.

The authors should also develop the last part of the introduction, leading to the objectives. How are we supposed to understand and accept the intervention proposed by the authors, if we have no more than one sentence and their non-supported statement to foster it? I mean, reading from line 59 (A novel meta-analysis…) on, it is not really clear why “therefore, exercise programs with different activities” could be the solution. Based on what? Is there any study, even not directly related to the topic, that supports this idea? The way the topic is presented, it sounds more like “we want to try and see what happens” rather than a statement built on solid reasoning/background

Our response: We have tried to improve and correct the mentioned statements in the introduction part. We hope that now seems clearer and understandable.

Lines 42-43: who supports the idea that there is a need of exercise programs? No citation is given, and the information in the introduction does not clarify this issue

Our response: Sorry for omitting this. We have added and change parts regarding the importance of exercise programs.

The authors say that: data is limited on programs for PF and health in preschool (lines 43-44, not supported by citation); there are several studies on exercise programs for fitness in preschool (lines 49-50, followed by two studies); there are limited studies investigating exercise programs in preschool (no citation). This is very confusing: several and limited are almost opposed concepts. Also, if studies are limited, what is the focus of the 19 studies in the meta-analysis reported? 19 studies perhaps are not many, yet it is already a body of research. Why not reporting the direct sources and what they found in those studies, rather than only mentioning this meta-analysis? How does the meta-analysis match the previous statements on limited/several studies?

Our response: Sorry for confusing introduction. We have tried to correct as suggested. Thank you for detailed comments. We really appreciate that.

Additionally, I would also suggest working on the logical sequence of topics presented so to give them order. Currently, the authors discuss (a) the importance of physical fitness in youth; (b) they declare there is a need for exercise programs to improve PF, with no supporting reference; (c) childcare importance for supporting active living; (d) they declare there are several studies on PA programs in pre-school (which contradicts their previous statement on the lack of data on preschool); (e) they switch to early childhood as an important age, mentioning that there is a lack of studies on it (few lines above, they stated the opposite – “several studies…” etc.); (f) finally, they mention that a meta-analysis showed that exercising has small effect on PF in preschoolers…I feel it is quite chaotic and difficult to follow, as well as contradicting itself in several points

Our response: As mentioned above. We agree that introduction seems confusing and hard to understand. We hope that we have improved it and corrected it so it would be suitable for further process.

Materials and Methods

Participants:

the sample size stated here (56) is inconsistent with that in the abstract (74) and with the sum of EG (38) and CG (36). Given that no information is provided about any participant dropping off, I assume it is a typo and needs to be corrected

Our response: We apologize for the inconvenience, this was a typographical error. There were 74 children included in this study.

Perhaps saying that they “volunteered” is not fully correct, as they are aged 5-6. It is quite difficult for me to imagine a 5-year-old kid saying he/she accepts to volunteer for a study, just as much as I find hard to imagine any kid of that age refusing to participate in some fun activity. I might be wrong, but I believe it would be more precise to say “whose parents gave permission” (as stated later) or similar

Our response: Indeed, kids aren’t independent. We agree with you; we did not use precise language. As a matter of fact, their parents gave informed consent (lines 68-70), therefore, the sentence has been removed.

Were all kids from one kindergarten? From more? If from more, were there differences among them (rural/urban, socioeconomic level of the neighbourhood, facilities available, etc.)? Those could be variables affecting the results, especially for the CG. Was the CG sample selected from the same kindergarten/s? Were they selected proportionally? A bit more information here would be helpful

Our response: We have mentioned as the study limitation that groups were not randomised. However, the control group was randomly chosen from same kindergardens as the children from MSG. We have added some additional information regarding the control group selection.

Table 1 is misplaced, why presenting results on pre-post anthropometric measures in the “participants” section?

Our response: You are right. Please find it in the Results section.

Testing procedures

What is the 20m dash testing? Speed?

Our response: You are right. 20m dash is primarily testing speed but also anaerobic power. We added in following manuscript sentence the purpose of the test.

Can the authors please report the reference/s for the validity data they mention in the section, please? It would be even better if the authors could report validity data in samples of the same age for each of the test separately (adding reference), rather than summing it up at the end with no cited work to support it.

Our response:  Thank you for your insight. We accepted your suggestion and we added missing references to all tests. Furthermore, we clarified some test protocols and added missing test – the medicine ball throw. Also, we revised and reorganized the list of the tests according to suggestions of one of the reviewers.

Multisport program

Line 126: I think the authors meant “trained” rather than “educated”

Our response: You are right, we replace it.

I am a bit confused, perhaps because the authors use the word “treatment” for both EG and CG. So, the EG was assisted by trained PE teachers. On line 145, the authors say that “the treatment” was conducted by non-experts. I assume they refer to CG, therefore, I would suggest not to use the same term (treatment) for both groups so to reduce confusion. “Treatment” is typically associated with an intervention (EG), whereas CG simply followed the regular activities they were supposed to do (no treatment).

Our response: We agree with you. It was improperly used term to describe Serbian regular activities in kindergartens, and we brought ambiguity. We substituted it with the //regular activities in kindergarten// So please find revised paragraph in the new version the manuscript.

Is the program based on any literature? Is there any specific reason why it was built this way? Perhaps background literature, experts’ feedback, etc.? It would be great to understand the reasoning in this; maybe, if the authors can improve the introduction and justify better the need for implementing multiactivity programs, this would be come unnecessary here

Our response: This is the program that has showed good results in past several years. However, this results were not published in high impact factor journals. Therefore, this was the opportunity to present the significant results that this program and intervention has on physical fitness in children. We have already showed significant improvement in motor competence following this program. However, it was interesting to see how this program affect the physical fitness in preschool children.

What is the difference between “work with stations” and “circuit work”? Since the authors do not explain it in the text but mention it in the table, it would be nice to have more information (at least personally, for me to understand).

Our response:  Thank you for your comment. Please find the detailed explanation of the different organizing model of exercising in new version of the manuscript.

Statistical analysis

Please report the reference work for the threshold values of η2

Our response: We apologize, the reference was neglected. Please find it in the new version of the manuscript.

Results

Please correct the word “extend”. It is, in fact, “extent”

Our response: Thank You! Please accept our apologies, it was made by mistake.

Information in table 3 should be placed in a more standard way. There is too much text, whereas tables are made to avoid this. Also, the format is not consistent (for instance, why is there a border under “Speed”, yet not under any other test?)

Our response: Please accept our apology for the inconvenience. We denoted the simple main effect of group and time with symbols (†, *), while the interaction effect has been left intact. Kindly, find revised Table 3 in the new version of the manuscript.

Discussion

Why is BMI discussed? It seems to be not included anywhere except when it is reported as a table under “participants”. Either the authors include BMI (or body shape) as one of the objectives, they show the data in the results, they analyze the data as well, etc., or they should remove the information. Other would be if they had focused on health-related physical fitness, which usually includes body composition/shape. However, the flaws of the introduction do not help in this case. Had we had a better presentation of the research, perhaps the inclusion of BMI would be understood better

Our response: We completely agree with you, our goal wasn’t exploring intervention effects on body composition, so the statement: ‘’ In contrast to the enhanced physical performance, body composition remained nearly unchanged.’’ has been removed.

So, were the instructors trained PE teachers or PhD holders with a focus on sports and sciences? Or both? The information is not consistent, please unify it.

Our response: Sorry for mistake in explaining this important part of study design. Researchers in exercise science have designed and supervised the program. PE teachers trained the children instructed by researchers.

Lines 213-214: this sentence sounds more fitting in the conclusions rather than in the middle of the discussion

Our response: Thank you for your suggestion. The sentence was moved in the discussion part.

Line 246: I think “speculate” is not a proper term to use. I would rather use “assume”, or “postulate”, “theorize”, etc.

Our response: Thank You! We replace it accordingly.

Lines 247-249: again, this would be more a “conclusion” sentence. However, the authors should not go beyond the boundaries of their study: they have not studied outdoors activities, and they only theorize that the lack of growth in CRF is due to the indoor nature of their proposed program. Therefore, reaching the conclusion that outdoor activity should be proposed cannot be justified

Our response: We have erased the requested speculation due to the fact that this was not the scope of our research. Thank you for your suggestion.

Why, in the authors’ opinion, has grip strength increased more in CG? This should also be discussed, I believe

Our response: We have provided the reference with the norms for grip strength which showed lower values in grip strength in control group compared to MSG which could be the reason for smaller improvements in MSG.

The nature of the study is experimental. In the specific case, I am not sure about the meaningfulness of the results for a main reason: adding two hours of sports per week in kids’ routine can only improve their fitness compared to those who do not have such possibility. Especially in kids, whose development is triggered by several activities, I consider this a limitation. I think it is logical that if a kid can play 120 minutes more per week for nine months, compared to those who do not add that activity time, the fitness development is faster. Different would be if there had been another EG with a different set of activities to compare the multisport program with. So, all in all, what we can really say here is that extra 120 minutes/week of exercising per week seems to be better than “no extra activity”…

The above stated is made even more difficult considering that: the extra 120 minutes of activities did not improve CRF more than it did in CG, which did not have those 120-minute activities. The conclusion, basically, is that if a kid plays multisports 2h per week, and another does not do anything active in those 2h, CRF will increase similarly (there are plenty of mediating variables that I am not considering here, this is just for the purpose of clarifying my thought). So are multisports really good? Similar logical thought can be applied to grip strength, which increased more in kids who did not do anything rather than those who played multisports…the only thing I can think about is whether the kids in EG were not following the regular exercise time (kindergarten requirements, lines 142-144)…? No mention is given in the text, nor what requirements are applied to kindergarten in the community in which the study was carried out (2h/week? 1h/week?). Perhaps being more precise in describing these aspects would help better understand the meaningfulness of the results, regardless of their magnitude and statistical significance

Our response: As mentioned above, randomised trial would show the significace of this programs in preschool children. However, this was hard to do in this age groups. Nevertheless, this program showed much better results in motor competence compared to same control group. However, this results were submitted in other journal. The improvement in motor competence and expecialy in gross motor coordination is expected in this age, while physical fitness is still not investigated enough in order to make some bigger conclusions and statements. We hope that the results from the present project that include randomised groups will show the importance of multisport programs in preschool children for the improvement of physical fitness and physical activity.

While the discussion could delve more on potential causes of the results found, it is acceptable. However, in my opinion, there are several concepts that belong to the conclusions.

Our response: We have changed some parts of discussion and also moved some sentences from discussion to conclusion.

Final comment:

After reading the text, the definition “longitudinal study” given in the abstract is not correct. This is a pre-post study lasting 9 months. Not only the duration of a study defines the design, but also frequency of measures etc. Longitudinal studies suppose tracking the same sample over a long time with repeated measures, which is not the case. I suggest the authors remove this definition as it is misleading for readers

Our response: We are sorry for this mistake. The term “longitudinal study” was the only term used in our country for all designs that included initial and final measurements. We have changed the term accordingly.

A moderate review of the language is suggested

Our response: Thank you for your suggestion. We have used gramarly and native speaker has edited the manuscript.

We appreciate all the reviewer’s positive and encouraging comments.

Round 2

Reviewer 3 Report

Abstract.

Line 17: I think that “aimed at determining” would be better

Introduction.

Line 36: please remove “the” before “health”

Line 47-48: the sentence “Moreover…settings” is not clear. Do the authors want to say that kids who spend more time at childcares are more sedentary than those kids who spend less time at childcares? Or that kids are more sedentary during childcare hours rather than when they are outside? The way the sentence is posed makes the statement ambiguous in this sense, please reformulate

Line 55-56: there is a missing preposition in this sentence “…multicomponent training programs XX preschool…”. Perhaps “for”, or “in”…?

I appreciate the efforts of the authors in enhancing their introduction and increasing the supporting literature

The structured multisport exercise program.

Lines 168-171: not really sure a reference to a study with older kids (7-8 vs. 5-6 in this study) may help supporting the validity of using RPE…is there any study with the same age group? Was the Borg scale adapted/graphical? Many kids in preschool have not yet learnt to read, so how would the classical numbered scale work? Perhaps someone explaining it to them, or giving them examples…? Kids’ perceived effort may be very tricky as they do not fully understand their body yet. Also, when they are involved in fun activities, they tend to “forget” their tiredness. To summarize, perhaps its use does not make much sense, unless better justified for the specific age of the study. This does not seem to affect the quality of the work carried out by the authors, as RPE seems only a secondary, “monitoring” value; therefore, unless the authors overcome the issues I mentioned, I would suggest to remove it.

Table 1.

Ok, now I see what RPE was used for. I think intensity could be important to monitor, yet I maintain the doubts I have regarding using RPE in such young kids (unless more suitable reference is provided)

Results.

Lines 227-228 and following: I think this sentence should be worked on a little. In English it is a bit odd to say “significantly more increased”. Perhaps something like “A significantly higher increase has been observed in the…etc.”. Adjectives such us “great” (you have it few lines later), or “large” are more commonly used than what you have in those line

Lines 235-236: in order to increase readability and for clearness purposes, I would remove “there has also…that”. A simpler and clearer way to say that is “MBT test performance significantly improved to a similar extent for both MSG and CG (….)”

I apologize for adding this now, especially after the authors made several changes to this section. However, I feel like the result section in general is too chaotic. I had to navigate several times from one point to another to find out where the information for a certain test was placed. Considering that the authors already report the majority of their results in an extensive table, and given the current difficulty in reading the text due to repetitiveness ( many consecutive sentences start with “a group-by-time interaction showed…”) and a perhaps excessive amount of numbers (that is what tables are for, to avoid putting all numbers in the text), I would suggest a complete (and rather simpler) way to report results in the text. I am aware that sometimes it is difficult to coordinate suggestions from different reviewers, which at times are even “opposite to each other”, but I think clarity of the results is essential. Some suggestions: we do not have to repeat every time that “the group-by-time interaction…”; why not to state it in the beginning and then only mention the major results? Leave to the table the work of showing detailed results. For instance:

Main effect of time showed that MSG improved significantly in all studied variables. Regarding CG, improvements were found for the majority of the variables studied, except for X (no changes) and X (significant decline; p = X); the group-by-time interaction showed that MSG had significantly larger increases in XX (p = X), XX (p = X), and XX (p = X) compared to CG. However, CG improved significantly more than MSG in X (p = X). Table X presents detailed information on the results from the analysis.

Again, just a suggestion, but I feel it would highly increase the readability of this section.

Discussion.

Why do the authors consider the improvement in cardiorespiratory fitness as “smaller”? They report significant improvements in both groups. Was the effect size for the pre-post intragroup comparison low? I do not think they report it… as far as we know from the results, CRF improved significantly. Unless extra data is reported, the interpretation can only be “there was improvement/there was not”. Smaller or larger cannot be really said, it is significant and that is all (again, unless specific effect size is reported for the pre-post comparison…I can only see group-by-time interaction effect size in the work). The absence of significant differences between CG and MSG at post-test does not indicate small improvements, rather than both improved similarly (as you also say in the result section”). Meaning that, both the “regular activities” and the MS program worked well for CRF (although we need to take into account also external factors)

Line 276: please remove “compared to CRF”. In case, it should be “opposed to CRF”, or “differently than in CRF”, or similar. However, it is better to remove it completely

Lines 283 and on: I do not believe you can really say that the musculoskeletal improvements were better than those in CRF…based on what? If based only on the fact that in CRF there was no difference at post-test between MSG and CG, then I believe it is not a proper statement. Both groups improved significantly. Why not focusing only on musculoskeletal changes in this portion of the discussion, without constantly getting back to making a not-really-meaningful comparison with CRF results?

Line 284: I apologize, but I still do not see why the fact that the activities were indoor should lead to lower CRF…how would you support this statement? There is no supporting citation for it. If you are convinced that indoor activities had a negative effect on CRF, you should delve a bit more on explaining it, or at least tell readers which literature you get this idea from. As for now, it seems just a random idea dropped there. However, repeating myself (sorry), this whole portion of the discussion is based on the wrong idea (or not properly explained) that musculoskeletal fitness improved more than CRF, and that CRF did not improve much…

Line 294: I think the authors should make a major job in building their discussion in a more logically sequenced way. The first part of the discussion “forced” the idea (not supported by the results) that CRF improvements were small and worse than those from musculoskeletal fitness. The focus seems to go from CRF to the latter, yet, here, again, the authors start discussing CRF again, like they had not done it before. In a certain way, they are even contradicting themselves (first part of the discussion works on explaining why CRF did not improve sufficiently, this part mentions -in line with the study findings- that CRF improved significantly).

Perhaps it is because of my approach to manuscript writing, but I would prefer to see a more structured presentation of the discussion of the results.

Line 298: why were MSG CRF improvements “smaller” than those in CG? The results show something completely different, i.e., improvements were statistically the same (as authors themselves state in the results section). So how can “same improvements” be smaller in one group than the other? It does not make sense to me. Additionally, even assuming that this conjecture was in line with the findings, it should be justified: why would it be logical that high baseline values of CRF led to lower improvements (not saying it is not true; indeed, some literature hints at this, but you would need to report that literature in the text so for readers to know that your statement is supported)

Lines 305 and on: the authors here try to justify the reason why “indoor activities” would be worse for the development of CRF. However, they report authors saying that CRF is developed by activities that, in fact, can also be developed indoor, especially through games. Walking and running can be easily performed in indoor games (see programs such as SPARK, just as an example). The point of CRF is not running straight for several km, but to keep a full body effort for a prolonged time. Proposing indoor games with certain rules would easily achieve that. Excluding swimming (requires specific indoor facilities) and cycling (static bicycles do not make much sense in toddlers), also jumping rope can be perfectly performed indoor. So, the assumption that indoor activities led to smaller CRF improvements in quite incorrect, in my opinion. However, we go back to my original statement: how can we say that CRF had smaller improvements compared to other fitness variables and compared to the CG group? None of the results suggests this

Round 3

Reviewer 3 Report

N/A